# Topical Application of OJI-204 Alleviates Skin Dryness, Dry Skin-Induced Itch, and Mechanical Alloknesis

**DOI:** 10.3390/biomedicines13030556

**Published:** 2025-02-21

**Authors:** Sumika Toyama, Tomoya Nakamura, Mitsutoshi Tominaga, Kenji Takamori

**Affiliations:** 1Juntendo Itch Research Center (JIRC), Institute for Environmental and Gender-Specific Medicine, Graduate School of Medicine, Juntendo University, 2-1-1, Tomioka, Urayasu-shi 279-0021, Chiba, Japan; su-toyama@juntendo.ac.jp (S.T.); nakamura10mo88@oji-gr.com (T.N.); tominaga@juntendo.ac.jp (M.T.); 2Oji Pharma Co., Ltd., 1-10-6, Shinonome, Koto-ku, Tokyo 135-8558, Japan; 3Department of Dermatology, Juntendo University Urayasu Hospital, 2-1-1, Tomioka, Urayasu-shi 279-0021, Chiba, Japan

**Keywords:** alloknesis, dry skin, itch, moisturization, pentosan polysulfate sodium, sulfated hemicellulose

## Abstract

**Background/Objectives**: The skin is an important barrier that protects against invasion by foreign substances and retains water in the body. Several skin diseases involve dry skin due to a disrupted skin barrier, and most skin diseases that appear on dry skin are accompanied by itch. Dry skin-induced itch and mechanical alloknesis reduce quality of life. Sulfated hemicellulose (i.e., pentosan polysulfate sodium), similar to heparin, is a compound belonging to the sulfated polysaccharide family; however, in contrast to heparin, it is derived from plant materials. We herein investigate the effects of the topical application of OJI-204, a sulfated hemicellulose made by purifying and chemically synthesizing hemicellulose, on dry skin in a mouse model. **Methods**: The mouse model of dry skin was generated using a mixture of acetone and ether with water. Either OJI-204 (3% or 10%) or 0.3% heparinoid, PBS (control), was applied twice a day to the **a**cetone and diethyl **e**ther/**w**ater (AEW)-treated area. The degree of skin dryness was evaluated by measuring transepidermal water loss and stratum corneum hydration. Scratching behavior was recorded the day before AEW treatment and the day after the final day, and an alloknesis assay was performed on the day after the final day. **Results**: We found that 3% or 10% OJI-204 attenuated dry skin conditions (erythema/hemorrhage, scarring/dryness, edema, and excoriation/erosion) and itchiness more effectively than 0.3% heparinoid. Furthermore, the degree of dryness improved to the same degree as that with heparinoid. OJI-204 also significantly reduced dry skin-induced spontaneous itch and mechanical alloknesis. **Conclusions**: These results suggest the potential of OJI-204 as a therapeutic or preventive agent for dry skin.

## 1. Introduction

As the body’s largest organ, the skin acts as the first physiological barrier to the external environment, preventing the entry of foreign substances, including harmful microorganisms and irritants, and is important for retaining body moisture [1,2,3]. Environmental and genetic factors, such as pH, humidity, temperature changes, and sunburn, may weaken the skin barrier and cause skin dryness [4,5,6,7]. Dry skin is characterized by a scaly, rough, cracked, and fissured surface, and is closely associated with the somatosensory sensation of itch [3]. In common dermatoses, such as xerosis, a decline in barrier function often parallels with increases in the severity of clinical symptomatology, including pruritus [8]. The phenomenon of itch hypersensitivity, which is caused by normally innocuous mechanical stimuli, is referred to as mechanical alloknesis (m-alloknesis) and has been reported in various mouse models and patients with dry skin-based skin diseases, such as xerosis [9]. Since pruritus leads to disease burden, the development of treatment and prevention methods that focus on the transition of disease states and conditions is needed.

About 50% of timber is cellulose, approximately 25% is hemicellulose, and the remaining 25% consists of lignin and other substances [10]. Cellulose has been extensively used to produce paper and fiber. In contrast, hemicellulose has primarily been used as fuel in factory boilers. We are focusing on the potential of hemicellulose and are developing drugs for the future. Oji Pharma has successfully synthesized OJI-204 (Figure 1), which is a compound categorized as sulfated hemicellulose, similar to heparinoids using original techniques.

Similar to heparinoid, sulfated hemicellulose including pentosan polysulfate sodium (PPS) is a compound that belongs to the sulfated polysaccharide family. PPS has been shown to exhibit anti-coagulant and anti-inflammatory activities [11,12,13]. However, it currently remains unclear whether sulfated hemicellulose exerts the same moisturizing effect as heparinoid or if it has other effects on itch.

In the present study, we used a mouse model of dry skin to examine the degree of dryness and the effects of the topical application of OJI-204 on itch and m-alloknesis induced by dry skin. We herein demonstrate that topically applied OJI-204 exerted moisturizing effects and suppressed itch and m-alloknesis.

## 2. Materials and Methods

### 2.1. Animals

Male C57BL/6J mice were purchased from SLC Japan (Shizuoka, Japan) at 6–7 weeks old and were subsequently kept in-house and used in the experiments at 8 weeks old. They were maintained in the experimental animal facility of Juntendo University Graduate School of Medicine under a 12 h light/dark cycle at a regulated temperature of 22 to 24 °C, with food and tap water provided ad libitum. All experiments on animals were approved by the Animal Ethics Committee at Juntendo University Graduate School of Medicine.

### 2.2. Treatment for Cutaneous Barrier Disruption

The protocol for cutaneous barrier disruption was performed as previously described with some modifications [14]. In brief, three days before the test, hair was removed with an electric shaver. OJI-204 (Oji Pharma Co., Ltd., Tokyo, Japan; Figure 1) was dissolved in sterilized PBS. Mice were topically treated with 3% or 10% OJI-204, PBS alone, or 0.3% heparinoid (HR; positive control) twice daily for 7 days 1 h after the AEW treatment. Under sevoflurane (Maruishi Pharmaceutical Co., Ltd., Osaka, Japan) anesthesia, cotton (2 × 2 cm) soaked in a mixture of acetone and diethyl ether (1:1) was placed on the shaved area for 15 s and replaced within 5 s with cotton of the same size soaked in distilled water for 30 s (AEW treatment). These treatments were performed twice daily for 7 consecutive days (Figure 2). The second AEW treatment was performed 6 h after the first OJI-204 application.

### 2.3. Evaluation of Skin Condition and Dermatitis

Transepidermal water loss (TEWL) and stratum corneum (SC) water content in treated areas were measured using Tewameter^®^ TM210 and Corneometer^®^ CM825 (Courage & Khazawa, Cologne, Germany), respectively, on the day before the AEW treatment began and on the third and final day of treatment [15]. To confirm the skin condition, skin inflammation was analyzed using clinical skin scores on the same day as TEWL and SC hydration measurements as previously described [15]. Briefly, the severity of dry skin was assessed according to the following four symptoms: erythema/hemorrhage, scarring/dryness, edema, and excoriation/erosion, with each being graded according to their presence and severity from 0 to 3 (none, 0; mild, 1; moderate, 2; severe, 3). The clinical skin score was defined as the sum of individual scores and ranged from 0 to 12.

### 2.4. Evaluation of OJI-204 for Scratching Behavior and Sedative Effects

Itch-related scratching behavior was analyzed as previously described with a slight modification. On the day before the AEW treatment and on the final day of treatment, mice (4 mice per observation) were acclimated in acrylic cages (19.5 × 24 × 35 cm) for at least 1 hr. Scratching behavior and locomotion activity were monitored for 12 h using an SCLABA^®^-Next system and SCLABA^®^-Next tracking software (Noveltec, Kobe, Japan, http://www.noveltec.jp/english/sclabanext-e.html, accessed on 19 February 2025), respectively.

### 2.5. M-Alloknesis Assay

The m-alloknesis assay was performed as previously described with some modifications [16]. The day after the final AEW treatment, each mouse was placed in a new cage and habituated for at least 1 h. Mice received three separate innocuous mechanical stimuli delivered using a von Frey filament (bending force: 0.07 or 0.16 g, Bioseb, Chaville, France) at separate, randomly selected sites within the shaven area. Each mouse received innocuous mechanical stimuli to the shaven area of the rostral back three times using the filament at intervals >5 s (average 20 s). Within a 7.5 min interval, this sequence was repeated 10 times (30 stimulations in total). M-alloknesis scores were calculated as the total number of scratching responses. The m-alloknesis assay was performed the day before AEW treatment was started and groups were randomized and assigned using stratified randomization.

### 2.6. Statistical Analyses

All statistical analyses were performed using GraphPad Prism 7 software (GraphPad Software, San Diego, CA, USA). Data were expressed as mean values ± SEM. Differences between groups were examined for significance by one-way ANOVA with Sidak’s multiple-comparison test and two-way ANOVA with Tukey’s multiple-comparison test or the Kruskal–Wallis test followed by Dunn’s multiple-comparisons test.

## 3. Results

### 3.1. Effects of OJI-204 on Dry Skin Scores, TEWL, and SC Hydration

The AEW treatment induced dry skin symptoms, such as scaling, when repeated each day (PBS (vehicle) group; average 0.85 ± 0.076) (Figure 3A,E). Although dry skin symptoms were observed in both the 0.3% HR and OJI-204 groups, they were less severe in the 0.3% HR group (average 0.55 ± 0.092) than in the vehicle group (Figure 3B,E), and the application of 3% (average 0.25 ± 0.077) or 10% OJI-204 (average 0.2667 ± 0.047) attenuated symptoms more than 0.3% HR (Figure 3C–E). Skin dryness and skin barrier disruption are characterized by increased TEWL and decreased SC hydration, respectively [17]. Therefore, we measured these parameters in murine skin after the AEW treatment. On day 3 of the AEW treatment, while TEWL increased in the other groups, this increase was significantly suppressed in the 3% OJI-204 group (average 17.66 ± 2.802). On day 7 of the AEW treatment, TEWL was slightly less in the 0.3% HR group (average 18.31 ± 2.099) than in the vehicle group (average 25.5 ± 0.926) (Figure 3F). On the other hand, TEWL was significantly less in the OJI-204 group (10% OJI-204 group average 15 ± 1.931) than in the vehicle group (Figure 3F). The decrease in SC hydration was significantly less in the 3% OJI-204 group (average 8.475 ± 0.350) than in the vehicle group (average 6.758 ± 0.273) on day 3, but did not significantly differ between the HR (average 6.85 ± 0.257) and OJI-204 groups (10% OJI-204 group average 7.958 ± 0.355) and the vehicle group on day 7 (Figure 3G).

### 3.2. Effects of the Topical Application of OJI-204 on Spontaneous Itch and m-Alloknesis

We examined the effects of the topical application of OJI-204 on spontaneous itch and m-alloknesis in AEW-treated mice (Figure 4). The number of scratching bouts was significantly higher in AEW-treated mice (vehicle group; average 566.75 ± 64.529) than before the AEW treatment (average 296.92 ± 17.049). No significant difference was observed between the 10% OJI-204 (average 445.5 ± 37.873) and 0.3% HR groups (average 454.42 ± 40.186) and the vehicle group; however, the number of scratching bouts was significantly lower in the 3% OJI-204 group (average 354.75 ± 44.064) than in the vehicle group (Figure 4A). In addition, HR (average 1627 ± 44.064) and OJI-204 (3%, average 5202 ± 2738.129; 10%, average 1307 ± 550.434) did not affect locomotion activity in mice (the vehicle group; average 1942 ± 418.499) (Figure 4B).

The number of scratching bouts induced by von Frey filaments at a low bending force (0.07 g) (m-alloknesis score) was significantly lower in the 3% OJI-204 group (average 1.333 ± 0.256) than in the vehicle group (average 4.417 ± 0.874) (Figure 4C). Furthermore, the number of scratching bouts (m-alloknesis score) induced by von Frey filaments at a stronger bending force (0.16 g) was significantly lower in the 3% (average 2.333 ± 0.527) and 10% OJI-204 groups (average 2.417 ± 0.529) than in the vehicle group (average 5.75 ± 0.760) (Figure 4D).

## 4. Discussion

Heparinoid, a mucosulfated oligosaccharide, is commonly used in medicine to moisturize and promote circulation. It is also used to treat inflammatory diseases, progressive palmoplantar keratoderma, sebum deficiency (senile xeroderma), frostbite, hypertrophic scars, and keloids [18,19]. Heparinoid is derived from the organs of livestock animals, such as pigs and cows; however, the use of vegetable-based raw materials is expected to replace animal-based sources in the future due to reasons such as reduced risk of zoonotic diseases and a lower environmental impact. Therefore, we synthesized a plant-derived sulfated hemicellulose called OJI-204 and examined its effect on dry skin.

The present results showed that the topical administration of OJI-204 significantly attenuated dry skin symptoms and suppressed spontaneous itch and m-alloknesis in a mouse model of dry skin (Figure 3 and Figure 4). In addition, the results obtained revealed that OJI-204 exerted moisturizing effects on dry skin (Figure 3).

Previous studies have reported that dry skin induces itch and m-alloknesis [20], while moisturizing and film dressing protection attenuated both itch and m-alloknesis [21,22,23]. Itch dysesthesias represent abnormal sensory neuronal states in which considerable itch is evoked, for instance by light cutaneous stimuli, such as from clothing (m-alloknesis), or where increased itch is perceived in response to stimuli that normally evoke no itch (hyperkinesis) [24]. Dry skin also promotes the elongation of cutaneous sensory nerve fibers, which partially contributes to the induction of dry skin itch and hypersensitivity [17,25]. It has been reported that protecting the skin with a film dressing can suppress the elongation of these sensory nerves [23]. Therefore, topical OJI-204 may have reduced dry skin itch and m-alloknesis by normalizing dry skin and suppressing itch and itch hypersensitizing of dry skin factors.

Another possible mechanism is the inhibition of IL-4 by OJI-204. Recent studies using neutralizing antibodies against IL-4, despite IL-4 levels being below the detection limit (pg/mL), have demonstrated its involvement in m-alloknesis in mice with dry skin [20,26]. Additionally, IL-4 has been reported to contribute to itch sensitization [27]. PPS has also been shown to exert biological effects such as inhibiting both blood coagulation and IL-4 [11,12,13]. These findings suggest that OJI-204 may reduce dry skin-induced itch and m-alloknesis by inhibiting IL-4-induced neuronal sensitization. In the present study, we focused solely on the medicinal effects of OJI-204. However, to further clarify how OJI-204 reduces spontaneous itch and m-alloknesis, we will investigate immunostaining of nerve fibers and structural changes in the skin in future research.

## 5. Conclusions

The present results suggest that OJI-204 has potential to prevent skin dryness, dry skin-induced spontaneous itch, and m-alloknesis, as well as to serve as a plant-derived moisturizer alternative to 0.3% heparinoid, although it did not affect SC hydration. OJI-204 is expected to have future clinical applications as an alternative to heparin analogues, providing skin moisturization and itch suppression for dry skin.

## Figures and Tables

**Figure 1 biomedicines-13-00556-f001:**
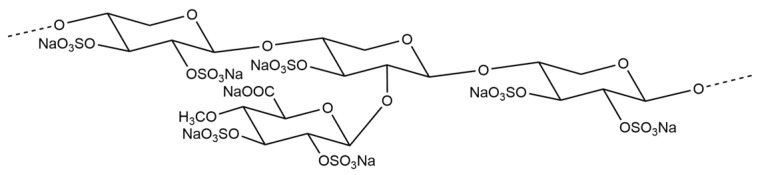
Chemical structure of OJI-204.

**Figure 2 biomedicines-13-00556-f002:**
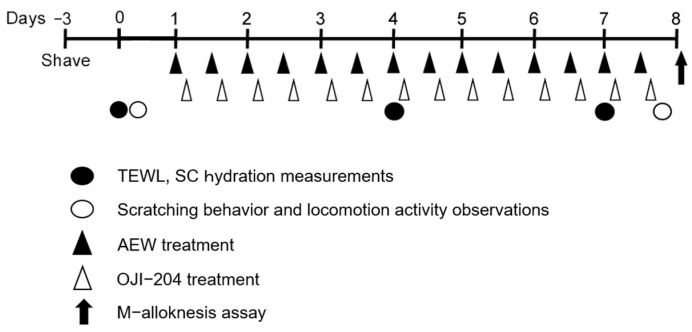
Experimental scheme.

**Figure 3 biomedicines-13-00556-f003:**
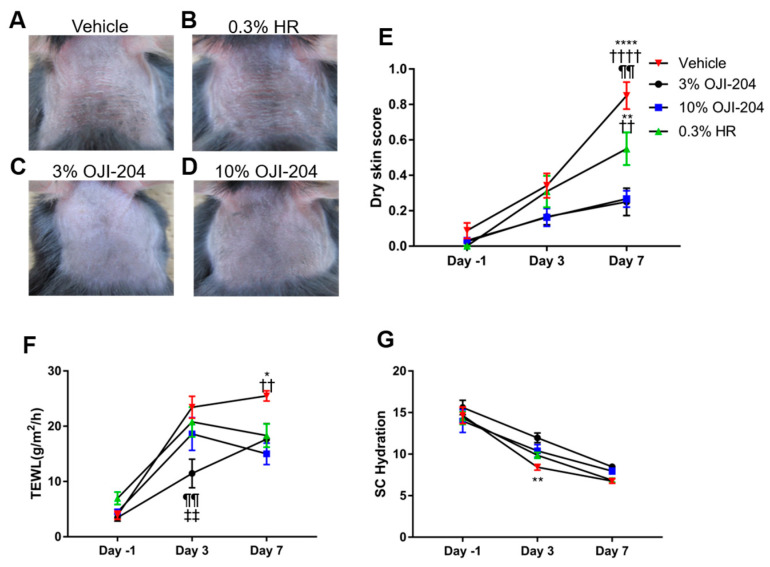
Effects of OJI-204 on dry skin. (**A**) Representative images of the back skin of AEW + PBS-treated mice after 7 days (upper left panels); (**B**) AEW + 0.3% HR-treated mice after 7 days (upper right panels); (**C**) AEW + 3% OJI-204-treated mice (lower left panels); and (**D**) AEW + 10% OJI-204-treated mice after 7 days (lower right panels). (**E**) Effects of treatment on dermatitis scores. (**F**) Effects of treatment on TEWL. (**G**) Effects of treatment on SC hydration. N = 12. * *p* < 0.05, ** *p* < 0.01, and **** *p* < 0.0001 significantly differed from AEW + 3% OJI-204-treated mice; ^††^ *p* < 0.01 and ^††††^
*p* < 0.0001 significantly differed from AEW + 10% OJI-204-treated mice; ^¶¶^
*p* < 0.01 significantly differed from AEW + 0.3% HR-treated mice, and ^‡‡^
*p* < 0.01 significantly differed from AEW +PBS-treated mice by two-way ANOVA with Tukey’s multiple-comparison test. The experiment was performed with N = 4. Numbers represent the means ± SEM of three independent experiments.

**Figure 4 biomedicines-13-00556-f004:**
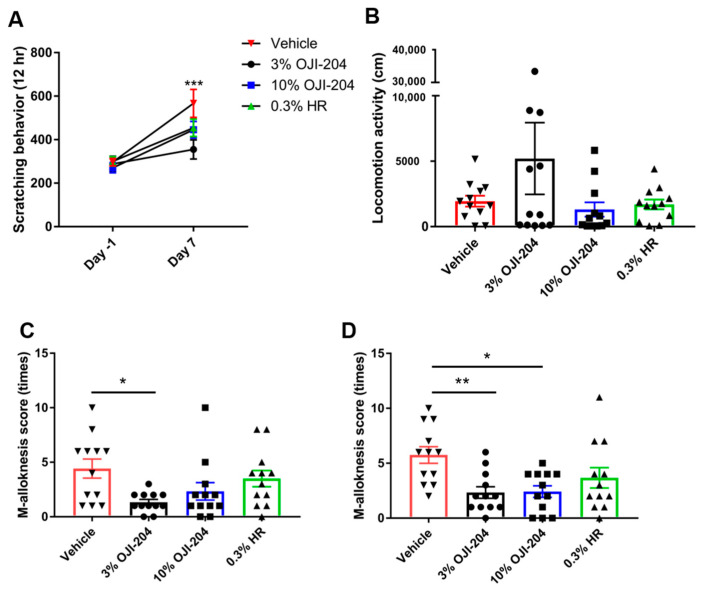
Effects of OJI-204 on itch. (**A**) Number of scratching behaviors. *** *p* < 0.001 significantly differed from AEW + 3% OJI-204-treated mice by two-way ANOVA with Tukey’s multiple-comparison test. (**B**) Locomotor activity. (**C**) Number of scratching bouts induced by von Frey filament (m-alloknesis score) stimulation with a bending force of 0.07 g or (**D**) 0.16 g after the AEW treatment in mice treated with 3% or 10% OJI-204, 0.3% HR, or PBS. N = 12. * *p* < 0.05 and ** *p* < 0.01 significantly differed from AEW + PBS-treated mice by one-way ANOVA with Sidak’s multiple-comparison test. The experiment was performed with N = 4. Numbers represent the means ± SEM of three independent experiments.

## Data Availability

All data are contained within the manuscript.

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
