# Peer review of "Topical Application of OJI-204 Alleviates Skin Dryness, Dry Skin-Induced Itch, and Mechanical Alloknesis"

_biomedicines, 2025, doi:10.3390/biomedicines13030556_

Round 1
Reviewer 1 Report
Comments and Suggestions for Authors
The communication-type article is interesting and well written, being easy to understand. There is also coherence between the methodology used, the results obtained and the discussion. However, there are two aspects that I consider would help improve the manuscript. 1. In the introduction section it would be important to include a more detailed state of the art and chronological order. In the conclusions section it is important to indicate the "negative" results and protections to be carried out in future studies.
Author Response
Comments and Suggestions for Authors
The communication-type article is interesting and well written, being easy to understand. There is also coherence between the methodology used, the results obtained and the discussion. However, there are two aspects that I consider would help improve the manuscript.
Response
Thank you for your comments and suggestions.
- In the introduction section it would be important to include a more detailed state of the art and chronological order. In the conclusions section it is important to indicate the "negative" results and protections to be carried out in future studies.
Response
Thank you for your comments. We have revised the Introduction and conclusions as follows:
Page 2 line 49-56 (Introduction)
About 50ï¼… of timber is cellulose, approximately 25ï¼… is hemicellulose, and the remaining 25ï¼… consists of lignin and other substances [10]. Cellulose has been extensively used to produce paper and fiber. In contrast, hemicellulose has primarily been used as fuel in factory boilers. We are focusing on the potential of hemicellulose and are developing drugs for the future. Oji Pharma has successfully synthesized OJI-204 (Fig. 1), which is a compound categorized as sulfated hemicellulose, similar to heparinoids by original technique.
Page 6 line 207-211 (Conclusions)
The present results suggest that OJI-204 has potential for preventing of skin dryness, dry skin-induced spontaneous itch, and m-alloknesis, as well as serving as a plant-derived moisturizer alternative to 0.3% heparinoid, although it did not affect SC hydration. OJI-204 is expected to have future clinical applications as an alternative to heparin analogues, providing skin moisturization and itch suppression for dry skin.

Reviewer 2 Report
Comments and Suggestions for Authors
This study explores the topical application of pentosan polysulfate sodium (PPS, OJI-204) as a treatment for skin dryness, itch, and mechanical alloknesis using a dry skin mouse model. PPS, a plant-derived polysaccharide similar to heparin, was tested at different concentrations (3% and 10%) to assess its effects compared to 0.3% heparinoid, a known moisturizing agent. While I appreciate this article, I think it could be improved in several ways. The study primarily focuses on IL-4, neglecting other cytokines or neural pathways that might also contribute to the itch response. Additionally, figures such as Figure 3 and Figure 4, which display significant markers, do not provide sufficient explanations for the changes observed under each condition or their potential causes. The absence of histological or immunohistochemical validation, such as histological sections or marker staining, limits the ability to demonstrate changes in skin structure or nerve fibers. The introduction includes lengthy or unnecessary background information that could be streamlined to better underscore the primary research problem. The Discussion section is too brief, particularly concerning skin inflammation. More narrative around wound inflammation research would be beneficial, as it's important to recognize recent biological advancements in the study of skin inflammation. For instance, during inflammatory responses like wounds, there's an increase in macrophage numbers, especially CX3CR1 bone-marrow-derived macrophages (Wang, X., Nat Commun 2017), which release TNF and TGFβ1 (Rahmani, W., J Invest Dermatol 2018). TNF, through the AKT/β-catenin signaling axis, stimulates hair follicle stem cells (HFSCs), leading to wound-induced hair anagen re-entry/growth (WIHA) and wound-induced hair follicle neogenesis (WIHN). The TGFβ1 signal plays a critical role in WIHA/WIHN and may initiate hair follicle regeneration via the AKT/PI3K pathway (Chen, H., PLoS One 2017; Chen, H., Theranostics 2019). The infundibulum, where hair follicles engage with external environments, is noted. Resident Langerhans cells, found in the upper section of the hair follicle, are crucial for overseeing the follicular canal and detecting potential pathogen invasions. Furthermore, as the infundibulum hosts various microbes, the local immune cell population may be influenced by the microbiome of the hair follicles (Moresi JM, J Cutan Pathol. 1997; Jaworsky C, Br J Dermatol. 1992). It is reported that upper HFSCs and the filling of sebaceous glands (SGs) (Han, J., iScience 2023) play significant roles. HFSCs and SG cells within these glands can further develop into sebaceous cells, contributing positively to wound healing (Han, J., iScience 2023). Finally, the conclusion is overly simplistic, failing to sufficiently highlight the practical application potential of OJI-204 or suggest directions for future research.
Comments on the Quality of English LanguageThe English could be improved to more clearly express the research.
Author Response
Comments and Suggestions for Authors
This study explores the topical application of pentosan polysulfate sodium (PPS, OJI-204) as a treatment for skin dryness, itch, and mechanical alloknesis using a dry skin mouse model. PPS, a plant-derived polysaccharide similar to heparin, was tested at different concentrations (3% and 10%) to assess its effects compared to 0.3% heparinoid, a known moisturizing agent. While I appreciate this article, I think it could be improved in several ways. The study primarily focuses on IL-4, neglecting other cytokines or neural pathways that might also contribute to the itch response. Additionally, figures such as Figure 3 and Figure 4, which display significant markers, do not provide sufficient explanations for the changes observed under each condition or their potential causes. The absence of histological or immunohistochemical validation, such as histological sections or marker staining, limits the ability to demonstrate changes in skin structure or nerve fibers. The introduction includes lengthy or unnecessary background information that could be streamlined to better underscore the primary research problem. The Discussion section is too brief, particularly concerning skin inflammation. More narrative around wound inflammation research would be beneficial, as it's important to recognize recent biological advancements in the study of skin inflammation. For instance, during inflammatory responses like wounds, there's an increase in macrophage numbers, especially CX3CR1 bone-marrow-derived macrophages (Wang, X., Nat Commun 2017), which release TNF and TGFβ1 (Rahmani, W., J Invest Dermatol 2018). TNF, through the AKT/β-catenin signaling axis, stimulates hair follicle stem cells (HFSCs), leading to wound-induced hair anagen re-entry/growth (WIHA) and wound-induced hair follicle neogenesis (WIHN). The TGFβ1 signal plays a critical role in WIHA/WIHN and may initiate hair follicle regeneration via the AKT/PI3K pathway (Chen, H., PLoS One 2017; Chen, H., Theranostics 2019). The infundibulum, where hair follicles engage with external environments, is noted. Resident Langerhans cells, found in the upper section of the hair follicle, are crucial for overseeing the follicular canal and detecting potential pathogen invasions. Furthermore, as the infundibulum hosts various microbes, the local immune cell population may be influenced by the microbiome of the hair follicles (Moresi JM, J Cutan Pathol. 1997; Jaworsky C, Br J Dermatol. 1992). It is reported that upper HFSCs and the filling of sebaceous glands (SGs) (Han, J., iScience 2023) play significant roles. HFSCs and SG cells within these glands can further develop into sebaceous cells, contributing positively to wound healing (Han, J., iScience 2023). Finally, the conclusion is overly simplistic, failing to sufficiently highlight the practical application potential of OJI-204 or suggest directions for future research.
Response
Thank you for your comments and suggestions. As you pointed out, cytokines other than IL-4, as well as neural factors, play a role in itch, particularly given the nerve fluctuations in AEW-dry skin.
To address these points, we have expanded the Discussion as follows:
Page 6 line 190–202 (Discussion)
Dry skin also promotes the elongation of cutaneous sensory nerve fibers, which partially contributes to the induction of dry skin itch and hypersensitivity [17, 25]. It has been reported that protecting the skin with a film dressing can suppress the elongation of these sensory nerves[23]. Therefore, topical OJI-204 may have reduced dry skin itch and m-alloknesis by normalizing dry skin and suppressing itch and itch hypersensitizing of dry skin factors.
Another possible mechanism is the inhibition of IL-4 by OJI-204. Recent studies using neutralizing antibodies against IL-4, despite IL-4 levels being below the detection limit (pg/ml), have demonstrated its involvement in m-alloknesis in dry skin mice[20, 26]. Additionally, IL-4 has been reported to contribute to itch sensitization [27]. PPS has also been shown to exert biological effects such as inhibiting both blood coagulation and IL-4[11-13]. These findings suggest that OJI-204 may reduce dry skin-induced itch and m-alloknesis by inhibiting IL-4-induced neuronal sensitization.
OJI-204 is a new drug candidate under development, and we received a response from the compound provider that they could not submit any reference to cytokines other than IL-4 due to trade secrets. Therefore, all comments in the discussion of inflammation other than IL-4 have been removed.
As you mentioned, the mechanism by which spontaneous itch and alloknesis are reduced can be partially elucidated through staining for specific markers. However, our study focuses solely on the medicinal effects of OJI-204, and we have submitted this article as a Communication to present these effects as quickly as possible. Therefore, elucidating the mechanisms of our findings remains a subject for future research.
Additionally, we have added the following to the Discussion:
Page 6 line 202–205(Discussion)
In the present study, we focused solely on the medicinal effects of OJI-204. However, to further clarify how OJI-204 reduces spontaneous itch and m-alloknesis, we will investigate immunostaining of nerve fibers and structural changes in the skin in future research.
You appear to equate wound healing with dry skin; however, dry skin is a precursor to inflammation rather than an active inflammatory state. As evident from the results of HE staining, inflammation is not occurring as you suggested.
Figure. Skin sections stained with H&E. (A)Vehicle (B)0.3% HR (C)3% OJI-204 (D)10% OJI-204. Scale bars = 100 μm.
The conclusion part has been modified as follows:
Page 6 line 207–211 (Conclusions)
The present results suggest that OJI-204 has potential for preventing of skin dryness, dry skin-induced spontaneous itch, and m-alloknesis, as well as serving as a plant-derived moisturizer alternative to 0.3% heparinoid, although it did not affect SC hydration. OJI-204 is expected to have future clinical applications as an alternative to heparin analogues, providing skin moisturization and itch suppression for dry skin.
Comments on the Quality of English Language
The English could be improved to more clearly express the research.
Response
Regarding this comment, another Reviewer noted that the communication-type article is both interesting and well written, making it easy to understand. If you found any specific phrases or sentences unclear, please let us know which ones. For your reference, we used the Medical English Service and David Price (Medical English service) for proofreading.

Reviewer 3 Report
Comments and Suggestions for Authors
In this communication, the author reported that the “Topical application of pentosan polysulfate sodium alleviates skin dryness, dry skin-induced itch, and mechanical alloknesis”.
1. Are there any significant differences between pentosan polysulfate sodium and OJI-204? If so, please explain it. If not, why is it named differently?
2. What is AEW? It is not discussed anywhere in the manuscript.
3. Is evaluating skin dryness using imaging techniques more appropriate than visual observations?
4. However, the presented results are too limited to understand the skin protection mechanisms of PPS.
Author Response
Comments and Suggestions for Authors
In this communication, the author reported that the “Topical application of pentosan polysulfate sodium alleviates skin dryness, dry skin-induced itch, and mechanical alloknesis”.
Response
Thank you for your comments and suggestions.
- Are there any significant differences between pentosan polysulfate sodium and OJI-204? If so, please explain it. If not, why is it named differently?
Response
Thank you for your question. Pentosan polysulfate sodium (PPS) and OJI-204 are classified as the same category of sulfated hemicellulose. OJI-204, in comparison to PPS (such as Elmiron®), exhibits a narrower molecular weight range and a higher side chain content, while maintaining similar pharmacological effects. Nonetheless, they are technically distinct compounds; hence, we have elected to distinguish between PPS and OJI-204 in this report.
(Due to confidentiality constraints, specific details regarding molecular weight and side chain content cannot be disclosed, as this information is proprietary to Oji Pharma.)
- What is AEW? It is not discussed anywhere in the manuscript.
Response
Thank you for your comment. AEW refers to acetone and diethyl ether/water treatment, which is described in the text as follows and also listed in the abbreviations section:
Under sevoflurane anesthesia (Maruishi Pharmaceutical Co., Ltd., Osaka, Japan), a cotton pad (2 cm × 2 cm) soaked in a 1:1 mixture of acetone and diethyl ether was placed on the shaved area for 15 seconds, followed within 5 seconds by a 30-second application of a cotton pad of the same size soaked in distilled water for 30 s (AEW treatment).
As you pointed out, this was unclear, so I have revised the Abstract for better clarity.
Page 1 line 22–23 (Abstract)
Either OJI-204 (3% or 10%) or 0.3% heparinoid, PBS (control), was applied twice a day to the acetone and diethyl ether/water (AEW)-treated area.
- Is evaluating skin dryness using imaging techniques more appropriate than visual observations?
Response
Thank you for your comment. As you mentioned, imaging techniques alone are not sufficient to evaluate the degree of skin dryness. However, imaging methods that capture wrinkles and texture can provide useful insights. In our study, we use the Tewameter and Corneometer to more accurately assess skin moisture content and the degree of dryness (Fig. 3).
- However, the presented results are too limited to understand the skin protection mechanisms of PPS.
Response
Thank you for your comment. As you pointed out, our study focuses primarily on the pharmacological efficacy of OJI-204. The underlying mechanisms by which it suppresses spontaneous itch and alloknesis remain unclear and will require further investigation.

Round 2
Reviewer 2 Report
Comments and Suggestions for Authors
Accept in present form
Reviewer 3 Report
Comments and Suggestions for Authors
The revised version may be considered for publication.